# Increasing the Passive Range of Joint Motion in Stroke Patients Using Botulinum Toxin: The Role of Pain Relief

**DOI:** 10.3390/toxins15050335

**Published:** 2023-05-13

**Authors:** Carlo Trompetto, Lucio Marinelli, Laura Mori, Nicola Bragazzi, Giulia Maggi, Filippo Cotellessa, Luca Puce, Lucilla Vestito, Franco Molteni, Giulio Gasperini, Nico Farina, Luciano Bissolotti, Francesco Sciarrini, Marzia Millevolte, Fabrizio Balestrieri, Domenico Antonio Restivo, Carmelo Chisari, Andrea Santamato, Alessandra Del Felice, Paolo Manganotti, Carlo Serrati, Antonio Currà

**Affiliations:** 1Department of Neuroscience, Rehabilitation, Ophthalmology, Genetics, Maternal and Child Health, University of Genoa, 16132 Genoa, GE, Italy; 2IRCCS Ospedale Policlinico San Martino, Division of Neurorehabilitation, Department of Neuroscience, 16132 Genoa, GE, Italy; 3IRCCS Ospedale Policlinico San Martino, Division of Clinical Neurophysiology, Department of Neuroscience, 16132 Genoa, GE, Italy; 4Laboratory for Industrial and Applied Mathematics (LIAM), Department of Mathematics and Statistics, York University, Toronto, ON M3J 1P3, Canada; 5Villa Beretta Rehabilitation Center, 23845 Costa Masnaga, LC, Italy; 6Rehabilitation Service, Fondazione Teresa Camplani Casa di Cura Domus Salutis, 25123 Brescia, BS, Italy; 7Intensive Rehabilitation Center, USL 1 Umbria, 06065 Passignano, PG, Italy; 8Clinica di Neuroriabilitazione, AOU Ospedali Riuniti, 60030 Ancona, AN, Italy; 9SOSD Gravi Cerebrolesioni Acquisite, AUSL Toscana Centro, 50141 Florence, FI, Italy; 10Neurologic Unit, Department of Medicine, “Garibaldi” Hospital, 95124 Catania, CT, Italy; 11Section of Neurorehabilitation, Department of Medical Specialties, University Hospital of Pisa, 56124 Pisa, PI, Italy; 12Spasticity and Movement Disorders “ReSTaRt” Unit, Physical Medicine and Rehabilitation Section, Policlinico Riuniti, University of Foggia, 71122 Foggia, FG, Italy; 13Department of Neuroscience, University of Padua, 35122 Padua, PD, Italy; 14Padua Neuroscience Center, University of Padua, 35122 Padua, PD, Italy; 15Clinical Unit of Neurology, Department of Medicine, Surgery and Health Sciences, Trieste University Hospital, University of Trieste, 34127 Trieste, TS, Italy; 16Department of Neurology, Imperia Hospital, 18100 Imperia, IM, Italy; 17Academic Neurology Unit, Department of Medico-surgical Sciences and Biotechnologies, Sapienza University of Rome, 04019 Terracina, LT, Italy

**Keywords:** spastic dystonia, spasticity, pathological postures, limb postures, stretch

## Abstract

By blocking the release of neurotransmitters, botulinum toxin A (BoNT-A) is an effective treatment for muscle over-activity and pain in stroke patients. BoNT-A has also been reported to increase passive range of motion (p-ROM), the decrease of which is mainly due to muscle shortening (i.e., muscle contracture). Although the mechanism of action of BoNT-A on p-ROM is far from understood, pain relief may be hypothesized to play a role. To test this hypothesis, a retrospective investigation of p-ROM and pain was conducted in post-stroke patients treated with BoNT-A for upper limb hypertonia. Among 70 stroke patients enrolled in the study, muscle tone (Modified Ashworth Scale), pathological postures, p-ROM, and pain during p-ROM assessment (Numeric Rating Scale, NRS) were investigated in elbow flexors (48 patients) and in finger flexors (64 patients), just before and 3–6 weeks after BoNT-A treatment. Before BoNT-A treatment, pathological postures of elbow flexion were found in all patients but one. A decreased elbow p-ROM was found in 18 patients (38%). Patients with decreased p-ROM had higher pain-NRS scores (5.08 ± 1.96, with a pain score ≥8 in 11% of cases) than patients with normal p-ROM (0.57 ± 1.36) (*p* < 0.001). Similarly, pathological postures of finger flexion were found in all patients but two. A decreased finger p-ROM was found in 14 patients (22%). Pain was more intense in the 14 patients with decreased p-ROM (8.43 ± 1.74, with a pain score ≥ 8 in 86% of cases) than in the 50 patients with normal p-ROM (0.98 ± 1.89) (*p* < 0.001). After BoNT-A treatment, muscle tone, pathological postures, and pain decreased in both elbow and finger flexors. In contrast, p-ROM increased only in finger flexors. The study discusses that pain plays a pivotal role in the increase in p-ROM observed after BoNT-A treatment.

## 1. Introduction

In stroke patients, muscles are hypertonic because of increased muscle activity during passive stretching, i.e., muscle over-activity [1,2,3], and/or because of decreased muscle extensibility caused by secondary muscle changes, including loss of sarcomeres and increased fat and collagen content [4,5,6,7].

Loss of sarcomeres in series along the myofibrils leads to muscle shortening. When the muscle shortens besides a certain threshold, the examiner cannot stretch it up to the end of passive range of motion (p-ROM). The ensuing decrease in p-ROM is called muscle contracture [8,9,10]. On the contrary, it is largely accepted that in stroke patients muscle over-activity does not cause p-ROM to decrease, since the examiner is expected to overcome most of the muscle over-activity during p-ROM assessment [11].

By blocking the release of acetylcholine at the neuromuscular junction, botulinum toxin A (BoNT-A) is an effective treatment for muscle over-activity [12]. In the upper limb of stroke patients, BoNT-A reduces muscle hypertonia [13,14,15], a finding expected because muscle over-activity is paramount for increasing muscle tone. Surprisingly, BoNT-A is also reported to increase p-ROM [16], a finding quite unexpected, because muscle contracture is the major determinant of decreased p-ROM [11].

In the literature, the mechanism leading BoNT-A to increase p-ROM is attributed to reducing muscle over-activity [16]. In stark contrast with the widespread notion that loss of p-ROM reflects muscle contracture, the interpretation reveals a real-world paradox, i.e., that muscle contracture is improved by BoNT-A action on muscle over-activity.

Having in mind the pain that the patient perceives during clinical examination helps to understand the apparent paradox. When evaluating patients with muscle hypertonia, the examiner must exert a force to complete the full range of p-ROM, especially in those with muscle shortening. If the patient reports pain, especially severe pain, the examiner is prompted to reduce the force impressed to the patient’s limb, thereby reducing p-ROM. Since BoNT-A has been proven to exert a potent analgesic effect on post-stroke muscles [17,18,19,20,21,22], we hypothesize that this analgesic effect may allow the examiner to impress greater force to the patient’s joint segment, thus increasing p-ROM in comparison to that assessed before the treatment.

To test this hypothesis, we conducted a retrospective investigation of p-ROM and pain in post-stroke patients treated with BoNT-A for upper limb hypertonia.

## 2. Results

Seventy patients met the inclusion criteria (age: 66 ± 11 years; sex: 25 women, 45 men; damaged hemisphere: right 38, left 32; lesion type: ischemia 47, haemorrhage 23; time since stroke onset: 7 ± 4 years; time interval between T0 and T1: 28 ± 3 days (range: 23–35 days)). Out of the 70 patients, 19 (27%) received oral drugs for spasticity: 13 patients received Baclofen (median dose = 50 mg), one received Baclofen 50 mg + Clonazepam 2 mg, one received Baclofen 50 mg + alprazolam 2 mg, one Pregabalin 225 mg, and three Gabapentin 200 mg. On the other hand, 22 patients (31%) received oral drugs for depression/pain: eight patients received Duloxetine 60 mg, seven Venlafaxine (median dose = 75 mg), two Fluoxetine 20 and 60 mg, four Sertraline 25 or 50 mg, and one Citalopram 20 mg.

Six patients had only elbow flexors injected, 22 patients only finger flexors, and 42 patients had both elbow and finger flexors injected. Thus, 48 patients were injected into elbow flexors and 64 patients into finger flexors (Table 1). BoNT-A was injected under ultrasound guidance. Muscles to be injected and toxin doses were determined in each single subject, according to the clinical picture, in the aim to reduce hypertonia, to improve pathological postures, and decrease the related disability and pain. After BoNT-A treatment, all patients underwent physiotherapy. No other adjunctive treatments were used.

### 2.1. Findings in Elbow Flexors before BoNT-A Injection (T0)

Among the 48 patients injected into elbow flexors, 30 patients (63%) had normal p-ROM (i.e., elbow angle of 0), while the remaining 18 patients (37%) had decreased p-ROM with an elbow angle >0 (28.89 ± 20.90). Sex, type of lesion, years since stroke, number of previous injections, and use of symptomatic treatment (for spasticity, pain and depression) were similar between patients with normal p-ROM and those with decreased p-ROM. In contrast, patients with decreased p-ROM were older than those with normal p-ROM (*p* = 0.017) (Table 2).

Patients with decreased p-ROM had higher MAS scores (2.94 ± 0.80) than those with normal p-ROM (1.70 ± 0.50) (*p* < 0.001).

With the sole exception of one patient with normal p-ROM, all patients had an elbow posture score >1 (i.e., pathological elbow posture). Elbow posture mean scores were 2.67 ± 0.49 in patients with decreased p-ROM and 2.30 ± 0.53 in those with normal p-ROM (*p* = 0.022).

Patients with decreased p-ROM had higher pain-NRS scores (5.08 ± 1.96) than patients with normal p-ROM (0.57 ± 1.36) (*p* < 0.001). All 18 patients with decreased p-ROM (100%) had a pain-NRS score >1; only 5 patients with normal p-ROM (17%) had a pain-NRS score >1. Among the 18 patients with decreased p-ROM, two (11%) had a pain-NRS score ≥8.

No differences of pain-NRS scores were found related to gender (*p* = 0.829).

### 2.2. Findings in Finger Flexors before BoNT-A Injection (T0)

Among the 64 patients injected into finger flexors, 50 patients (78%) had normal p-ROM (i.e., p-ROM score = 1), while the remaining 14 patients (22%) had decreased p-ROM (p-ROM score 3.57 ± 0.85). Age, sex, type of lesion, years since stroke, number of previous injections, and use of symptomatic treatments (for spasticity, pain, and depression) did not differ between groups.

MAS score was higher in the 14 patients with decreased p-ROM (3.57 ± 0.51) than in the 50 patients with normal p-ROM (2.54 ± 0.50) (*p* < 0.001).

Except for two patients with normal p-ROM, all patients had a finger posture score > 1 (i.e., pathologic finger posture). Finger posture score was higher in the 14 patients with decreased p-ROM (2.71 ± 0.47) than in the 50 patients with normal p-ROM (2.18 ± 0.48) (*p* = 0.001).

Pain was higher in the 14 patients with decreased p-ROM (8.43 ± 1.74) than in the 50 patients with normal p-ROM (0.98 ± 1.89) (*p* < 0.001). All 14 patients with decreased p-ROM (100%) had a pain-NRS score >1, while only 13 patients with normal p-ROM (26%) had a pain-NRS score >1. Among the 14 patients with decreased p-ROM, 12 patients (86%) had a pain-NRS score ≥8 (Table 2).

No differences of pain-NRS scores were found related to gender (*p* = 0.144).

### 2.3. Findings in Elbow Flexors after BoNT-A Injection (T1)

Figure 1 shows elbow p-ROM values at T0 and T1 (left panel) in the 18 patients with decreased p-ROM. These values did not differ between T0 (28.89 ± 20.90) and T1 (26.67 ± 20.79), *p* = 0.20.

In all injected patients (*n* = 48), muscle tone and elbow posture scores decreased at T1 (MAS: 2.17 ± 0.87 vs. 1.31 ± 1.01, *p* < 0.001; posture: 2.44 ± 0.54 vs. 1.80 ± 0.74, *p* = 0.001). Among patients showing pain at T0 (*n* = 23), pain decreased at T1 (4.72 ± 1.92 vs. 2.70 ± 2.44, *p* < 0.001) (Figure 2, top panels).

Changes in muscle tone, posture, and pain did not differ between patients with normal p-ROM and those with decreased p-ROM (respectively, *p* = 0.277, *p* = 0.730, *p* = 0.975) (Figure 2, bottom panels). Results remained consistent after adjusting for the use of treatments for spasticity, depression, and pain.

### 2.4. Findings in Finger Flexors after BoNT-A Injection (T1)

Figure 1 shows finger p-ROM scores at T0 and T1 (right panel) in the 14 patients with decreased p-ROM. The scores decreased at T1 (3.57 ± 0.85 vs. 2.50 ± 0.65) (*p* < 0.001).

In all injected patients (*n* = 64), muscle tone and posture scores significantly decreased at T1 (MAS: 2.77 ± 0.66 vs. 1.34 ± 1.18, *p* < 0.001; posture: 2.30 ± 0.52 vs. 1.84 ± 0.57, *p* = 0.009). Among patients showing pain at T0 (*n* = 27), pain decreased at T1 (6.19 ± 2.94 vs. 3.48 ± 3.52, *p* < 0.001) (Figure 3, top panels).

Only changes (before and after BoNT-A) in muscle tone differed between patients with normal p-ROM and those with decreased p-ROM (T0-T1 change: −1.60 ± 0.99 vs. −0.79 ± 0.89, *p* = 0.005), whereas changes in posture and pain did not differ (respectively, *p* = 0.561 and *p* = 0.200) (Figure 3, bottom panels).

## 3. Discussion

This retrospective study in post-stroke patients was designed to test whether BoNT-A treatment influences p-ROM of the elbow and fingers flexors, and to investigate whether this influence is exerted by BoNT-A action on pain.

We observed that most post-stroke hypertonic patients with normal p-ROM have no pain during passive muscle stretching. The minority who complains of pain (26% of patients with finger flexor hypertonia and 17% of patients with elbow flexor hypertonia), reported mild or moderate pain intensity (pain ≤ 5 in all but one subject with finger hypertonia).

In contrast, all patients with decreased p-ROM have pain. Patients with elbow flexor hypertonia and decreased p-ROM have mild to moderate pain, with a pain-NRS score ≥ 8 only in 11% of cases. Patients with finger flexor hypertonia and decreased p-ROM have markedly intense pain during muscle stretching, with a pain score ≥8 in 86% of cases.

As expected, after BoNT-A treatment, the MAS scores, posture scores, and pain-NRS scores decreased in both elbow and finger flexors; interestingly, p-ROM increased at fingers, but it remained unchanged at the elbow.

### 3.1. Spasticity or Spastic Dystonia?

In accordance with the inclusion criteria, all enrolled patients were affected by muscle hypertonia of the elbow flexors and/or finger flexors.

In patients with Upper Motor Neuron Syndrome (UMNS), muscle hypertonia can be determined by two different phenomena: spasticity and spastic dystonia [1,23,24]. Spasticity is the exaggeration of the stretch reflex [25], where the involuntary muscle activity is present only during the dynamic phase of passive muscle stretch and is velocity dependent: the higher the stretch velocity, the greater the involuntary muscle activity [26,27]. For very low stretch velocity, such as those used to assess p-ROM in this study, spasticity is usually not evoked [28]. Otherwise, spastic dystonia refers to the inability to voluntarily silent muscle activity on command [29]. The ensuing spontaneous tonic contractions are increased by muscle stretch. They lead to both length-dependent hypertonia and pathological postures, with the upper limb usually adducted and flexed [30,31].

In the last two decades, it has been proposed to use the term spasticity more extensively, to indicate all positive UMNS phenomena without further distinction of the different forms of muscle over-activity [32]. Although in certain contexts this view may prove useful and fully acceptable [33], in the present study the classic distinction between spasticity and spastic dystonia (for a review, see [31,34]) needs to be maintained, since spastic dystonia—but not spasticity—contributes to explain the findings of the present study.

The overwhelming majority of the patients enrolled had spastic dystonia, as they not only had muscle hypertonia, but also exhibited pathological postures. This finding is in line with the widely accepted view that abnormal upper limb postures in stroke patients are a disabling phenomenon that deserves BoNT-A treatment [35]. Since no patient with spasticity was enrolled, it follows that upper limb spasticity in stroke patients is a poorly disabling phenomenon, requiring no BoNT-A treatment.

That the examiner manages to overcome much of the muscle over-activity during p-ROM assessment in stroke patients is widely assumed [11]. However, this assumption has never been systematically explored nor has it been fixed. From a theoretical point of view, unlike spasticity, spastic dystonia may oppose the examiner maneuver, thereby decreasing p-ROM. In fact, spastic dystonia is present also when the muscle is stretched slowly, as we did to evaluate p-ROM. Moreover, its length-dependent nature causes spastic dystonia to increase at increasing muscle length [34], especially when the stretched muscle approaches the maximum range of p-ROM.

No matter how crude the measures used, in our patients BoNT-A treatment improved pathological postures by reducing both elbow and finger flexion. Although post-stroke patients with upper limb hypertonia treated with BoNT-A pathological postures have been rarely evaluated [16], our results suggest that a precise assessment of upper limb posture is likely to be the most accurate outcome measure to evaluate the clinical effects of BoNT-A.

### 3.2. Pain Pathophysiology in Patients with Spastic Dystonia and Mechanisms of BoNT-A Action on Pain

In a recent study, we discussed that the passive stretching of an actively contracted muscle provokes pain [20]. For example, it happens in healthy subjects who exert an eccentric muscle contraction when resisting a forced stretch [36]. Reasonably, the same is supposed to happen in spastic dystonia when assessing muscle tone and p-ROM [20]. Although previous studies have shown that gender plays a role in pain perception in stroke patients [37,38], in the present study no significant difference in pain-NRS scores was found between men and women, possibly due to sample size.

The present study shows that patients with decreased p-ROM always complain of pain during passive joint mobilization, with intensity that tends to be severe at fingers. Conversely, most patients with normal p-ROM have no pain during passive joint mobilization, or very mild pain if present. These results suggest that pain is likely due to remodeling that leads to muscle shortening. The same is also likely true in patients with normal p-ROM and pain, in whom muscle shortening is not sufficient to decrease p-ROM, yet able to induce pain due to the excessive mechanoreceptor activation during p-ROM assessment.

Pain is a highly disabling symptom in patients with spastic dystonia. The present study shows that it is not an intrinsic feature of spastic dystonia, rather a consequence of the muscle shortening frequently accompanying. This distinction is extremely important, as it implies that pain is not an unavoidable consequence of spastic dystonia, rather it can be—and should be—prevented by preventing muscle shortening, a goal recognized as feasible in acute and post-acute stroke patients [31].

Overall, pain reduction after BoNT-A treatment can be induced by the muscle relaxant effect acting on spastic dystonia and/or by a specific action along the nociceptive pathway acting on the sensory neurons [39]. Indeed, BoNT-A has been reported to reduce human mechanical pain sensitivity and mechano-transduction [40]. Because we collected pain data in patients with spastic dystonia and it decreased after treatment, we cannot establish whether the BoNT-A analgesic effect in our patients was mediated by waning spastic dystonia, by direct anti-nociceptive action, or both.

### 3.3. P-ROM Changes in Elbow and Finger Joints

Before treatment with BoNT-A, the 18 patients with decreased p-ROM of the elbow had spastic dystonia and pain. In theory, both phenomena could have contributed to limiting p-ROM. BoNT-A injection attenuated spastic dystonia, as documented by decreased muscle hypertonia and benefit on elbow flexion postures. In addition, BoNT-A injection also reduced pain (Figure 2). However, p-ROM remained unchanged (Figure 1), indicating that during the evaluation prior to BoNT-A, elbow p-ROM was limited neither by pain nor by spastic dystonia. We argue that p-ROM was limited by muscle shortening, known to be unaffected by BoNT-A.

Before treatment with BoNT-A, the 14 patients with decreased p-ROM of the fingers had spastic dystonia and pain. As with the elbow flexors, BoNT-A treatment both alleviated spastic dystonia and reduced pain (Figure 2), but unlike the case of elbow flexors, it also increased p-ROM (Figure 1). Thus, prior to BoNT-A treatment, pain, or spastic dystonia, or both could have contributed to limiting p-ROM.

As every clinician who examines post-stroke patients daily knows, the larger the muscles subjected to passive stretching, the greater the difficulty to overcome spastic dystonia [11]. It would be difficult to sustain the argument that the examiner was limited in completing p-ROM by spastic dystonia generated in smaller muscles (i.e., finger flexors) than that generated in larger muscles (i.e., elbow flexors). Therefore, we are inclined to reason that pain was precisely the factor limiting finger p-ROM prior to BoNT-A injection. Pain reduction after BoNT-A allowed the examiner to exert a greater force, thereby increasing finger p-ROM. Consistently, in the elbow flexors, where pain was less intense, no increase in p-ROM was reported. Furthermore, the finger joints are much smaller than the elbow, and the examiner, while evaluating the patient with pain, may have been more cautious with fingers than with elbow, considering the smaller joints more susceptible to damage. Overall, pain in the elbow was less intense than in fingers, and possibly the examiner was less concerned about damaging the elbow than the fingers during p-ROM assessment. Therefore, before injection, elbow pain would have not limited p-ROM and, consequently, pain relief by BoNT-A would not have increased p-ROM.

### 3.4. Limitation of the Study

This study has some limitations. First, some patients’ features are heterogeneous (age, time since stroke, number of previous BoNT-A injections), injection schemes were individualized, and sample size is limited. Second, it is a retrospective analysis of a single clinician, which may limit applicability and generalization of findings. Third, the study is inherent in its nature as a real-world clinical practice, in which toxin is administered by relying on conventional injection points, rather than on observed intramuscular patterns of nerve distribution [41,42], and in which muscle contracture is evaluated clinically, and not measured with instrumental devices (e.g., magnetic resonance imaging and ultrasound).

## 4. Conclusions

Despite the limitations reported beforehand, the findings from the present study allow to address some points. The real clinical target of upper-limb BoNT-A injections in post-stroke patients proves to be spastic dystonia, because it is a real disabling phenomenon, as confirmed by the fact that no patients with spasticity were enrolled. Pain is not an intrinsic feature of spastic dystonia, rather it is inherent to muscle shortening (frequently, but not always, accompanying spastic dystonia). Finally, presence of pain is crucial for the toxin to increase the p-ROM.

These points have practical consequences. In stroke patients with spastic dystonia of the upper limb, elbow and finger joint pain can be prevented, since preventing muscle shortening is a feasible goal in acute and sub-acute stroke patients. Pain reporting during muscle tone assessment is mandatory when p-ROM is used as a BoNT-A treatment outcome measure.

## 5. Materials and Methods

### 5.1. Study Design

We conceived the study protocol at a group scientific meeting focused on BoNT-A effects on pROM, hypertonia, and pain. Subsequently, we retrospectively reviewed the medical records of patients injected with BoNT-A (IncobotulinumtoxinA) for the treatment of muscle hypertonia of elbow and finger flexors between January 2019 and December 2022. The same examiner (CT) made all the assessments.

We focused on the elbow and finger flexors rather than on muscles acting on the shoulder, because most of the patients with hypertonic upper limb are infiltrated in elbow and finger flexors. Although shoulder pain is frequent in patients with upper limb hypertonia and most of the publications assessing the effect of BoNT-A on pain in stroke patients are focused on shoulder pain (Struyf et al., 2023), the muscles acting on this joint are infiltrated only in a minority of cases, at least in our patients.

The present study was conducted in accordance with the World Medical Association’s Code of Ethics (Declaration of Helsinki) for experiments involving humans. Written informed consent was obtained from all participants. The study was approved by the local ethics committee (N. Registro CER Liguria: 40/2023–DB id 12953).

### 5.2. Patients’ Selection

#### 5.2.1. Inclusion Criteria


Age ≥ 18 years;Chronic hemiparesis due to a single stroke occurred >6 months before the assessment;Muscle hypertonia of elbow and/or finger flexors treated with BoNT-A;Clinical assessment performed just before (T0) and 3–6 weeks after BoNT-A (T1) including: (a) extension p-ROM measurements of elbow and fingers; (b) pain perceived at the elbow and fingers during extension p-ROM measurement; (c) muscle tone of elbow and finger flexors; (d) pathological postures of elbow and fingers.


#### 5.2.2. Exclusion Criteria


Recurrent strokes;Other medical conditions in addition to stroke likely to interfere with the clinical assessment reported in the inclusion criteria;Use of intrathecal baclofen;BoNT-A injection in the upper limb in the three months before assessment;Severe cognitive impairment (score of Mini Mental State Examination < 21);Severe aphasia interfering with patient’s assessment.


### 5.3. Assessment of p-ROM

Extension p-ROM of elbow and fingers was assessed. During the evaluation, the patient was lying on a bed in the supine position. The patient was asked to stay completely relaxed. Muscle stretching of the flexor muscles was performed as slowly as possible to minimize the dynamic stretch reflex [43]. The stretching maneuver was performed by exerting a force aimed to overcome the resistance offered by muscle over-activity evoked by the stretch [11].

For elbow extension p-ROM measurement, the examiner moved the subject’s forearm from the most flexed to the most extended position. This passive movement was performed slowly and forcefully to the point where further passive extension was not possible or would cause severe pain (point of maximum stretch). The elbow angle at the point of maximum stretch was measured with a hand goniometer. When full elbow extension was reachable, it was recorded as 0, whereas positive values indicated p-ROM decrease.

For finger extension p-ROM measurement, the examiner moved the subject’s fingers from the most flexed to the most extended position. This passive movement was performed slowly and forcefully to the point where further passive extension was not possible or would cause severe pain (point of maximum stretch). Finger posture at the point of maximum stretch was rated according to the following five levels: (1) fingers fully extended (normal p-ROM); (2) fingers 75% extended (mild loss of p-ROM); (3) fingers 50% extended (moderate loss of p-ROM); (4) fingers 25% extended (severe loss of p-ROM); (5) fingers not extensible (full loss of p-ROM) [18].

### 5.4. Pain Assessment

Elbow and finger flexors pain during passive extension were rated by the patients by scoring the pain intensity on a vertical Numerical Rating Scale (pain-NRS) ranging from 0 to 10 [20,44].

### 5.5. Muscle Tone Assessment

Tone of elbow and finger flexors was evaluated using the Modified Ashworth Scale (MAS), a 6-point scale ranging from 0 (no increase in tone) to 4 (limb rigid in flexion or extension) [45]. In the analysis of MAS score, 1+ was transformed into 1.5.

### 5.6. Postures Assessment

Upper limb postures were assessed while the patient was walking or, if unable to walk, while standing alone or supported.

Elbow postures were scored into the following three categories: (1) elbow flexion < 135° (normal flexion); (2) elbow flexion between 135° and 90° (moderate hyperflexion); (3) elbow flexion > 90° (severe hyperflexion).

Finger postures were classified into the following three categories: (1) finger posture as the contralateral hand (normal); (2) fingers flexed more than the contralateral side, but no clenched fist (moderate hyperflexion); (3) clenched fist (severe hyperflexion).

### 5.7. Statistical Analysis

Results are shown as mean (standard deviation) for continuous or ordinal variables and as absolute frequency (percentage) for categorical variables. Baseline characteristics of the patients, respectively, included in the elbow and fingers evaluation were reported jointly as well as separately for patients with normal p-ROM and patients with decreased p-ROM. The two groups were compared using *t*-test for the continuous variables and Chi-square test or Fisher’s exact test for the categorical variables. P-ROM, muscle tone, and pain changes over time were evaluated performing linear mixed models with random intercept or mixed-effect logistic models, always adjusting for age and sex. Additionally, a sensitive analysis was performed adjusting for the use of symptomatic treatment for spasticity, pain, and depression. P-ROM was studied among patients with decreased p-ROM at baseline, pain was evaluated among patients reporting pain at baseline, whereas muscle tone and posture were assessed in all participants. To assess the differences between patients with normal and decreased p-ROM, the interaction between time and group was tested.

A two-sided α less than 0.05 was considered statistically significant. All statistical analyses were performed using Stata version 16.0 (Stata Corporation, College Station, TX, USA).

## Figures and Tables

**Figure 1 toxins-15-00335-f001:**
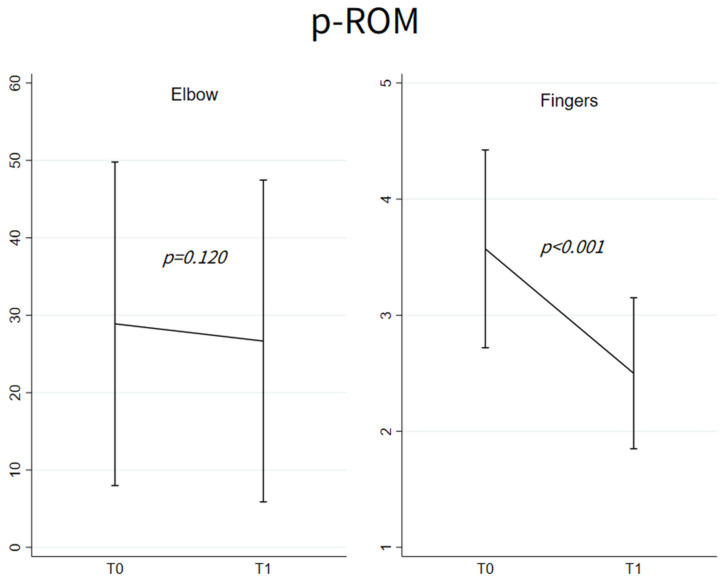
Mean ± SD of p-ROM at T0 and T1 among patients with abnormal p-ROM at T0, respectively, for elbow flexors (*n* = 18) and fingers flexors (*n* = 14); *p*-values refer to the evaluation of the change from T0 based on the linear mixed models with random intercept adjusted for age and sex.

**Figure 2 toxins-15-00335-f002:**
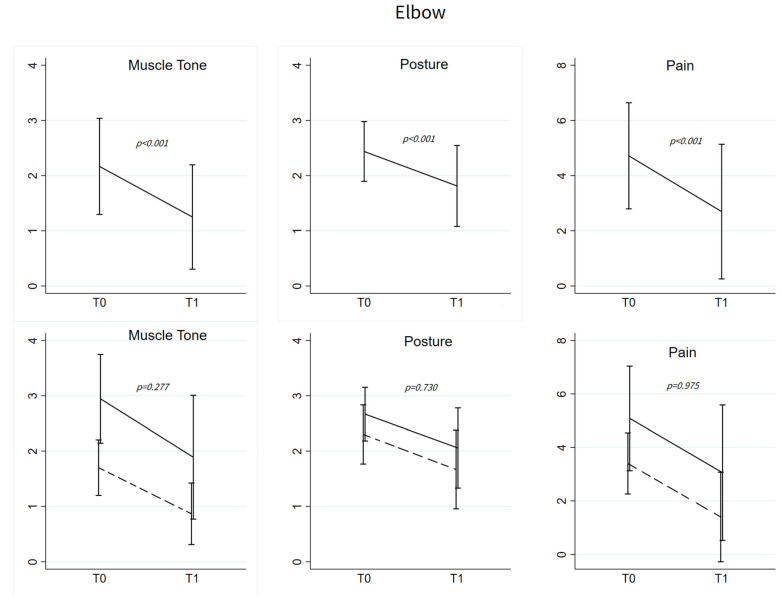
Elbow flexors. (**Upper**): Mean ± SD of muscle tone, posture and pain at T0 and T1; *p*-values refer to the evaluation of the change from T0 based on the linear mixed models with random intercept for muscle tone and pain and to mixed-effects ordered logistic regression model for posture scores. All the models were adjusted for age and sex. Muscle tone and posture T0-T1 changes were evaluated for all the patients (*n* = 48), while improvement in pain was studied only among patients with pain at T0 (*n* = 23). (**Below**): Mean ± SD of muscle tone, posture and pain at T0 and T1 among patients with normal p-ROM (dashed line) and abnormal p-ROM (solid line) at T0. *p*-values refer to the comparison of the pattern changes between the two groups testing the Time*Group interaction in linear mixed models or in the mixed-effects ordered logistic model.

**Figure 3 toxins-15-00335-f003:**
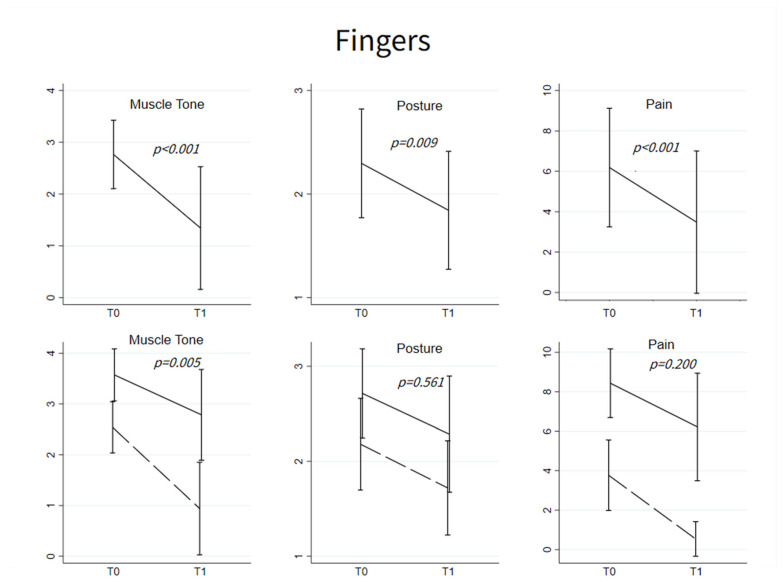
Fingers flexors. (**Upper**): Mean ± SD of muscle tone, posture and pain at T0 and T1; *p*-values refer to the evaluation of the change from T0, based on the linear mixed models with random intercept for muscle tone and pain and to mixed-effects logistic regression model for severe abnormal flexion in posture scores. All the models were adjusted for age and sex. Muscle tone and posture T0-T1 changes were evaluated for all the patients (*n* = 64), while improvement in pain was studied only among patients with pain at T0 (*n* = 27). (**Below**): Mean ± SD of muscle tone, posture and pain at T0 and T1 among patients with normal p-ROM (dashed line) and abnormal p-ROM (solid line) at T0. *p*-values refer to the comparison of the pattern changes between the two groups testing the Time*Group interaction in linear mixed models or in the mixed-effects logistic model.

**Table 1 toxins-15-00335-t001:** Muscles injected and doses of BoNT-A (incobotulinum A) for each muscle.

Muscle	Number of Injected Patients (Percentage)	Dose of BoNT-A: Mean ± SD (Range)
**Elbow flexor muscles** **(48 patients)**		
Biceps brachii	47/48 (98%)	82 ± 31 (50–150)
Brachialis	47/48 (98%)	63 ± 20 (50–100)
Brachioradialis	22/48 (46%)	49 ± 5 (25–50)
**Finger flexor muscles** **(64 patients)**		
Flexor digitorum superficialis	60/64 (94%)	108 ± 57 (25–300)
Flexor digitorum profundus	49/64 (77%)	70 ± 26 (25–150)
Intrinsic finger muscles	50/64 (78%)	67 ± 24 (25–100)

**Table 2 toxins-15-00335-t002:** Characteristics of the patients at T0 in all participants, and according to normal or decreased p-ROM. *p*-values refer to *t*-test for continuous variables and to Chi-square or Fisher’s exact test for categorical variables.

	All	Normal p-ROM	Decreased p-ROM	*p*-Value
ELBOW	*n* = 48	*n* = 30	*n* = 18	
Age, mean (SD)	66.44 (10.92)	63.57 (10.08)	71.22 (10.83)	**0.017**
Male Sex, n (%)	32 (67%)	22 (73%)	10 (56%)	0.206
Ischemic lesion, n (%)	35 (73%)	23 (77%)	12 (67%)	0.450
Years since stroke, mean (SD)	7.41 (4.60)	6.68 (4.23)	8.63 (5.04)	0.158
Number of previous injections, mean (SD)	7.13 (5.76)	7.27 (5.94)	6.89 (5.61)	0.829
Under treatment for spasticity	14 (29%)	10 (33%)	4 (22%)	0.521
Under treatment for depression	14 (29%)	10 (33%)	4 (22%)	0.521
MAS at T0	2.17 (0.87)	1.70 (0.50)	2.94 (0.80)	**<0.001**
Loss of p-ROM at T0	---	---	28.89 (20.90)	---
Posture at T0	2.44 (0.54)	2.30 (0.53)	2.67 (0.49)	**0.0216**
Pain at T0	2.26 (2.72)	0.57 (1.36)	5.08 (1.96)	**<0.001**
**FINGERS**	***n* = 64**	***n* = 50**	***n* = 14**	
Age, mean (SD)	65.36 (11.16)	65.14 (10.56)	66.14 (13.52)	0.769
Male Sex, n (%)	40 (63%)	33 (66%)	7 (50%)	0.274
Ischemic lesion, n (%)	43 (67%)	32 (64%)	11 (79%)	0.305
Years since stroke, mean (SD)	6.75 (4.46)	6.69 (4.42)	6.98 (4.75)	0.832
Number of previous injections, mean (SD)	5.97 (5.76)	5.90 (5.61)	6.21 (6.48)	0.858
Under treatment for spasticity	16 (25%)	12 (24%)	4 (29%)	0.736
Under treatment for depression	21 (33%)	14 (28%)	7 (50%)	0.196
MAS at T0	2.77 (0.66)	2.54 (0.50)	3.57 (0.51)	**<0.001**
Loss of p-ROM at T0	---	---	3.57 (0.85)	---
Posture at T0	2.30 (0.52)	2.18 (0.48)	2.71 (0.47)	**0.001**
Pain at T0	2.61 (3.61)	0.98 (1.89)	8.43 (1.74)	**<0.001**

## Data Availability

Data can be provided upon reasonable request.

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
