# Peer review of "Increasing the Passive Range of Joint Motion in Stroke Patients Using Botulinum Toxin: The Role of Pain Relief"

_toxins, 2023, doi:10.3390/toxins15050335_

Round 1

Reviewer 1 Report

Significant methodological issues. This is a retrospective review but would have been stronger as a  prospective case series . Also not enough information about patient demographics (ie. any previous pain symptoms, medications used, targeting of BoNTa (Emg, NCS, Ultrasound?), any adjunctive therapy- as this can all increase outcome and pain). Also there are significant limitations to this study (and the authors only mention 2 lines for their limitations- limitations is that this is a retrospective analysis of only one clinician, the study may already be biased because of only looking at the practice pattern of one clinician and its applicability is therefore limited. The manuscript is difficult to follow as they start to introduce new concepts outside the scope of what they are investigating- for example talking about spastic dystonia. At times this manuscript sounds like a narrative review rather than the intended retrospective chart analysis. The title is not clear- this is a single practice retrospective chart analysis. Also comments that all patients with p-ROM have pain needs to be better discussed as this is only based on this retrospective analysis and can also be secondary to increased force and pressure of stretch.

Reviewer 2 Report

This is a well-written manuscript where the authors assessed the effect of botulinum toxin on the increase of passive range of motion (p-ROM) in the upper limb in stroke patients and its link to pain relief. Overall, the report is well organized. Yet, some changes should be reconsidered.

Minor revisions:

·         Introduction:

·         Page 2, Line 50: “In stark contrast with the widespread notion that loss of p-ROM reflects muscle contracture, that interpretation reveals a real-world paradox, i.e., that muscle contracture is improved by BoNT-A action on muscle over-activity!”è avoid the exclamation mark  

·         Formulate the sentence at the end of the introduction as a hypothesis: “Since BoNT-A has been proven to exert a potent analgesic effect on post-stroke muscles (Shaw et al., 2011; Rosales et al., 2012; Wissel et al., 2021; Trompetto et al., 2022; Pavone 58 and Luvisetto, 2010; Wang et al., 2022), reasonably this analgesic effect allows the examiner to impress greater force to the patient's joint segment, thus increasing p-ROM in comparison to that assessed before the treatment.” è Since BoNT-A has been proven to exert a potent analgesic effect on post-stroke muscles (Shaw et al., 2011; Rosales et al., 2012; Wissel et al., 2021; Trompetto et al., 2022; Pavone and Luvisetto, 2010; Wang et al., 2022), we hypothesize that this analgesic effect may reasonably allow the examiner to impress greater force to the patient's joint segment, thus increasing p-ROM in comparison to that assessed before the treatment.

·         Methods

o   Is the study retrospective or prospective? Why did the authors retrospectively review patients’ charts? At which moment the study protocol was established and applied?

o   Page 2, line 83: Exclusion criteria: use numbering as inclusion criteria

·         Results

o   Has the effect of gender on pain perception been analyzed? Was there an adjustment according to gender? This should be added to the results and the discussion section as gender has been reported to play a role in pain perception in stroke patients (The Effects of Gender, Functional Condition, and ADL on Pressure Pain Threshold in Stroke Patients. Zhang YH, Wang YC, Hu GW, Ding XQ, Shen XH, Yang H, Rong JF, Wang XQ. Front Neurosci. 2021 Jul 30;15:705516. doi: 10.3389/fnins.2021.705516.)// Sex differences in the symptom presentation of stroke: A systematic review and meta-analysis. Shajahan S, Sun L, Harris K, Wang X, Sandset EC, Yu AY, Woodward M, Peters SA, Carcel C. Int J Stroke. 2023 Feb;18(2):144-153. doi: 10.1177/17474930221090133.)

·         Discussion

o   How do the authors explain the differences between the findings in the Elbow and the finger flexors in terms of p-ROM increase? This should be better detailed.

o   Most of publications assessing the effect of botulinum toxin on pain in stroke patients rather focused on shoulder pain (The Place of Botulinum Toxin in Spastic Hemiplegic Shoulder Pain after Stroke: A Scoping Review. Struyf P, Triccas LT, Schillebeeckx F, Struyf F. Int J Environ Res Public Health. 2023 Feb 4;20(4):2797. doi: 10.3390/ijerph20042797.). How do the authors explain the focus on finger and elbow and not wrist and shoulder?   

o   UMNS: abbreviation to be introduced (Upper Motor Neurons)

o   Page 10, line 308: it can produces pain due to the excessive mechanoreceptor activation during p-ROM assessmentè it can produce pain due to the excessive mechanoreceptor activation during p-ROM assessment

o   Page 11, line 314: it can be - and should be! - prevented by preventing muscle shortening, a goal recognized as feasible in acute and post-acute stroke patients è avoid the exclamation mark

o   The limitation section needs to be better formulated and developed

·         Conclusion

o   The conclusion section needs to be better formulated

Reviewer 3 Report

The study describes a retrospective investigation of post-stroke patients treated with botulinum toxin A (BoNT-A) for upper limb hypertonia. BoNT-A is known to be effective in treating muscle over-activity and pain in stroke patients, and it has also been reported to increase passive range of motion (p-ROM). The investigation examined muscle tone, pathological postures, p-ROM, and pain before and after BoNT-A treatment. The results showed that patients with decreased p-ROM had higher pain scores than patients with normal p-ROM before treatment, and after treatment, muscle tone, pathological postures, and pain decreased in both elbow and finger flexors.

In the abstract

In sentence 6, consider adding "muscle" before "shortening" for clarity.

In sentence 8, consider replacing "supposed" with "hypothesized" for stronger language.

In sentence 16, consider adding a comma after "scores" to separate the dependent and independent clauses.

In sentence 22, consider replacing "We discussed" with "The study discusses" for a more objective tone.

In the introduction

Please check line 51 “!”

adding a comma after "type of lesion" in line 158 and after "patients with normal p-ROM" in line 159.

In the method, please explain how has the musch units of BoNT has been used? Based on what? what kind of toxins were used? Is it onato? Type A?

In the discussion, please added the limitation of the study that it had conventional injection point rather than observing intramuscular patterns of the nerve distribution.  Distribution of the intramuscular innervation of the brachial triceps: clinical importance in the treatment of spasticity with botulinum neurotoxin” “Effective botulinum toxin injection guide for treatment of cervical dystonia

I would like to give major revision and want to have reviewed again after revised version.

Reviewer 4 Report

The authors dealt with e very intriguing subject: Stroke spasticity and spastic dystonia and more specifically the effect of BoNT-A on p-ROM.

There are some methodological issues that need to be taken care of before this research study is ready to be submitted.

1. The number of the participant is to small and just reporting it in the limitation does not change the fact that the study is underpowered to make conclusions about this matter.

2. The aforementioned is also supported by the fact that there are demographic differences that could may influence the results (eg age).

3. Furthermore on page 9 (4.1 Spasticity or spastic dystonia), whereas you mention the study of Marinelli 2017, you have taken the theory of spastic dystonia as the sole reason of hypertonia in stroke patients. This is not the fact as both spasticity and more recently discovered spastic dystonia occur in such patients and not only one of them. Thus, further on (linew 278-280), you report that you had no patient with spasticity, to justify your findings which is not right. It is known that spasticity is one of the core features of the stroke course and is connected with spastic dystonia.

4. On page 10 (4.3 P-ROM changes in elbow and fingers joints), you present many arguing that could support your findings and the difference between the elbow and finger joints. These are mere hypotheses and are strongly subjective. Especially in lines 330-331 where you argue that p-ROM was limited by muscle shortening, known to be unaffected by BoNT. Did you try to measure which muscles had shortening? Did you do MRI? Why this phenomenon is not present in the finger joints of your patients? There is a lot to discuss and to argue about this part.

5. There are more limitations and the authors should present them more extensively.

6. In the methods section the authors mention as an exclusion the administration of intrathecal baclofen. What about per os baclofen or tizanidine? Dosage? Duration? Was there a sub analysis?

7. The time duration of possible assessment is to wide. Patient in 3 weeks may not have the same effect as those assessed in 6 weeks.

For those reasons I strongly believe that the study needs more data and more clear methods to be scientific sound.

Round 2

Reviewer 3 Report

All the questions has been resolved. Thank you for the answers.

Reviewer 4 Report

Thank you for taking into consideration the comments and adding more patients to have a better estimation and higher power.